# Dog brains are sensitive to infant- and dog-directed prosody

Anna Gergely [1,6] ✉, Anna Gábor [2,3,6], Márta Gácsi[2,4], Anna Kis[1], Kálmán Czeibert[2], József Topál[1] & Attila Andics[2,3,5]

When addressing preverbal infants and family dogs, people tend to use specific speech styles. While recent studies suggest acoustic parallels between infant- and dog-directed speech, it is unclear whether dogs, like infants, show enhanced neural sensitivity to prosodic aspects of speech directed to them. Using functional magnetic resonance imaging on awake unrestrained dogs we identify two non-primary auditory regions, one that involve the ventralmost part of the left caudal Sylvian gyrus and the temporal pole and the other at the transition of the left caudal and rostral Sylvian gyrus, which respond more to naturalistic dog- and/or infant-directed speech than to adult-directed speech, especially when speak by female speakers. This activity increase is driven by sensitivity to fundamental frequency mean and variance resulting in positive modulatory effects of these acoustic parameters in both aforementioned non-primary auditory regions. These findings show that the dog auditory cortex, similarly to that of human infants, is sensitive to the acoustic properties of speech directed to non-speaking partners. This increased neuronal responsiveness to exaggerated prosody may be one reason why dogs outperform other animals when processing speech.

[1] Institute of Cognitive Neuroscience and Psychology, ELTE-ELKH NAP Comparative Ethology research group, Research Centre for Natural Sciences, Budapest, Hungary. [2] Department of Ethology, Eötvös Loránd University, Budapest, Hungary. [3] Neuroethology of Communication Lab, Department of Ethology, Eötvös Loránd University, Budapest, Hungary. [4] ELKH-ELTE Comparative Ethology Research Group, Budapest, Hungary. [5] ELTE NAP Canine Brain Research Group, Budapest, Hungary. [6] These authors contributed equally: Anna Gergely, Anna Gábor. ✉email: gergely.anna@ttk.hu

Exploring how companion animals are neurally prepared for how we speak to them is key to understanding how living with humans may have tuned ancient auditory mechanisms to process speech more efficiently. There is growing evidence that family dogs (*Canis familiaris*) are feasible candidates for such investigations, as they can be tested with non-invasive cognitive neuroscience methods[1], and are responsive to different cues carried by human speech (lexical, emotional prosodic, speaker's identity, word novelty; e.g.[2–7]).

When addressing family dogs, human speakers tend to utilize speech styles characterized by high and variable pitch and short utterances, similar to those used to address infants[8–10]. This suggests that the exaggerated prosody used toward dogs and infants possesses general acoustic characteristics to call and maintain the attention of a social partner with limited linguistic competence. The higher fundamental frequency (F0) mean in dog- and infant-directed speech (DDS and IDS) compared to adult directed speech (ADS) is characteristic of both male and female speakers. In addition to the well-documented and anatomy-based pitch and formant frequency differences in female and male speech[11,12], there is also evidence that genders tend to use their addressee-specific voice differently. Women hyper-articulate their vowels more in IDS than men[13], while no such difference can be found in DDS and ADS. Moreover, women generally use a wider pitch range than men for addressee-specific prosody[10].

Infants' behavioral and neural responses to addressee-specific speech prosody have already been widely investigated. In infant studies, natural IDS has been used as an acoustic stimulus. Compared to ADS, IDS has been characterized by heightened pitch, wider pitch range, greater pitch variability and more exaggerated vowels (e.g.[10,14]). The function of such addressing style is to make the child highly attentive and to enhance positive emotions during the interaction. This promotes secure bonding between participants, and also has a significant effect on infants' cognitive, social, and language development (reviewed in ref. [15]). Behavioral studies in infants indicate preference toward IDS over ADS. Interestingly, this preference is typically stronger for female speakers ([16,17] but see[18]). This difference in infants' reactions to speakers of different genders is usually explained by the extended intrauterine and postnatal exposure to female (i.e., the mother's) voice[19,20]. Although no infant fMRI study is available on the topic, other neuroimaging techniques provided insight into the neural machinery modulating IDS preference. EEG and fNIRS studies showed that preverbal infants' brains discriminate IDS from ADS of female but not of male speakers[21–23] and vowels of women are more easily distinguished by the infants when hearing them in IDS as opposed to ADS[24]. Moreover, increased bilateral activation was found in temporal areas of 4–13-month-old infants when they were presented with female IDS compared to female ADS[25]. These results clearly indicate that addressee-specific speech prosody and gender-specific acoustic cues therein have significant effects on infants' behavioral and neural responses (for a review see ref. [20]).

Infant-like behavioral preference has also been shown in dogs toward DDS over ADS of female speakers[9]. To date, however, no behavioral preference or different neural sensitivity has been shown in dogs in response to female versus male speech. Previous studies showed that dogs are sensitive to gender-related acoustic parameters of human speech[26] and different postnatal experiences with the two genders can have an effect on their behavior[27]. Moreover, similarly to IDS, speakers' gender affects DDS as women tend to talk more to dogs than men during natural playing situations and use more exaggerated prosody[28,29]. Importantly, however, no direct comparisons have been made between dogs' responses to naturally spoken DDS vs. IDS or

between dogs' responses to female vs. male DDS. The neural underpinnings of dogs processing addressee-specific characteristics of exaggerated prosody in speech also remain to be explored.

In the present paper we applied non-invasive fMRI on awake family dogs to investigate neural sensitivity to addressee-specific (DDS/IDS/ADS) prosody and whether such sensitivity is gender-specific (female/male speakers). Dogs were listening to natural speech samples collected from various female and male speakers unfamiliar to the dogs. Speakers were talking to their own dog (DDS) and preverbal infant (IDS), and another adult (ADS) during the recording (see ref. [10]). Because of the considerable acoustic differences between ADS and DDS/IDS, and the similar acoustics and attention-getting function of DDS and IDS[10,30], we expected distinct, increased auditory cortical responses to DDS/IDS compared to ADS, and similar responses to DDS and IDS. We also tested whether the dog brain, similarly to the infant brain, preferentially processes female DDS/IDS. Finally, we aimed to identify neural sensitivities to acoustic parameters relevant for addressee-specific prosody. Based on previous findings[9,30,31] we expected such sensitivities for fundamental frequency-related acoustic parameters (F0 mean, F0 variance, F0 change).

## Results

**Effects of addressee-specific prosody and speaker gender**. Neural sensitivity to the addressee (DDS, IDS, ADS) and the speaker's gender (female (F), male (M)) was evaluated using whole brain, GLM-based random-effects analyses (Table 1). This revealed robust clusters responsive to auditory conditions (DDS + IDS + ADS > Silence (Sil)) involving the bilateral primary auditory cortex (left and right middle ectosylvian gyrus: L/R mESG) and left auditory subcortical regions (a cluster involving parts of the caudal colliculus and the medial geniculate body: CC/MGB) and left non-primary auditory cortical regions (a cluster involving the most ventral part of the caudal Sylvian gyrus and the temporal pole: cSG/TP) (Table 1; Fig. 1). The inverse contrast (DDS + IDS + ADS < Sil) revealed activity in the left caudate nucleus (CN). We found no main or addressee-specific effects of speaker gender, but addressee, and addressee by gender interaction effects were found. Namely, a cluster at the transition of the left caudal and rostral (c/r) SG was more responsive to conditions with exaggerated prosody (i.e., DDS and IDS) especially when spoken by females. More specifically, increased L c/rSG response was revealed by DDS > ADS, FDDS > FADS, FIDS > FADS, FDDS + FIDS > FADS, and (FDDS + FIDS > FADS) > (MDDS + MIDS > MADS) contrasts (Table 1; Figs. 1, 2). The L cSG-TP also responded stronger to female speech with exaggerated prosody (FDDS + FIDS > FADS) (Table 1). We found no significant effects for any other contrasts including one-to-one and two-to-one comparisons, whether collapsed or separated by gender, nor for further Gender × Addressee interaction contrasts (see Table 1 for details).

Follow-up analysis using the cephalic index as covariate revealed no individual differences either in the whole brain, or within the relevant ROIs (L c/rSG, L cSG/TP; all cluster-corrected $P > 0.05$ for FWE) in the contrasts revealing significant effects in the previous analysis (see Table 1).

**Effect of acoustic parameters**. We assumed that the above effects are driven by the acoustic differences amongst the conditions, thus the next step was to identify key acoustic elements behind them. Detailed acoustic analysis of the auditory stimuli showed significant differences between addressee-specific prosody (i.e., DDS, IDS, and ADS) and/or gender of the speaker in all the acoustic parameters investigated (namely the F0 mean, variance and

**Table 1 Whole-brain random-effects tests of addressee and speaker's gender processing.**

| Contrast | Brain region | x | y | z | Cluster size | T12 | P_FWE cluster |
|---|---|---|---|---|---|---|---|
| **Auditory** | | | | | | | |
| All sounds > Sil | L mESG | −20 | −20 | 20 | 238 | 9.756 | <0.001 |
| | L cSG-TP | −22 | −12 | −2 | 40 | 6.913 | <0.001 |
| | L CC-MGB | −6 | −24 | 0 | 58 | 4.590 | <0.001 |
| | R mSSG-mESG | 22 | −24 | 18 | 215 | 4.557 | <0.001 |
| All sounds < Sil | L CN | −4 | 8 | 8 | 37 | 5.164 | 0.001 |
| **Gender** | No significant clusters for any contrasts | | | | | | |
| **Addressee** | | | | | | | |
| DDS > ADS | L c/rSG | −22 | −16 | 4 | 23 | 5.309 | <0.001 |
| FDDS > FADS | L c/rSG | −22 | −10 | 8 | 41 | 5.502 | <0.001 |
| FIDS > FADS | L c/rSG | −22 | −10 | 6 | 15 | 5.107 | 0.036 |
| FDDS + FIDS > FADS | L cSG-TP | −20 | −6 | −4 | 15 | 7.157 | <0.001 |
| | L c/rSG | −22 | −10 | 6 | 36 | 6.604 | <0.001 |
| | No significant clusters for any other contrasts | | | | | | |
| **Gender × Addressee** | | | | | | | |
| (FDDS + FIDS > FADS) > (MDDS + MIDS > MADS) | L c/rSG | −22 | −12 | 8 | 14 | 5.577 | 0.047 |
| | No significant clusters for any other contrasts | | | | | | |

Threshold for reporting all contrasts was $P < 0.001$ and whole-brain cluster-corrected $P < 0.05$ for FWE, clusters of minimum 3 voxels (24 mm³). Only the strongest peak is reported for each cluster. The following contrasts did not result in significant clusters: Gender: F > / < M, FDDS > / < MDDS, FIDS > / < MIDS, FADS > / < MADS; Addressee: DDS < ADS, FDDS < FADS, FIDS < FADS, FDDS + FIDS < FADS, IDS > / < ADS, DDS > / < IDS, FDDS > / < MIDS, MDDS > / < MADS, MIDS > / < MADS, FDDS + FADS > / < FIDS, FIDS + FADS > / < FDDS, MDDS + MIDS > / < MADS, MDDS + MADS > / < MIDS, MIDS + MADS > / < MDDS; Gender×Addressee: (FDDS + FIDS > FADS) < (MDDS + MIDS > MADS), (FDDS > FADS) >/< (MDDS > MADS).
*L* left, *R* right, *m* mid, *c* caudal, *CC* caudal colliculus, *CN* caudate nucleus, *c/rSG* caudo-rostral transition of the Sylvian gyrus, *ESG* ectosylvian gyrus, *MGB* medial geniculate body, *SG* Sylvian gyrus, *SSG* suprasylvian gyrus, *TP* temporal pole. *ADS* adult-directed speech, *DDS* dog-directed speech, *IDS* infant-directed speech, *F* female, *M* male, *Sil* silence, *n.s.* no significant clusters.

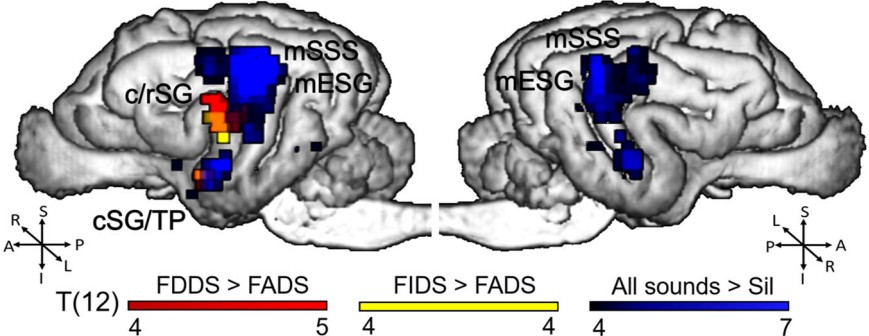

**Fig. 1 Dog brain responses to auditory stimuli.** This figure represents All sounds > Sil whole brain random effects test and FDDS > FADS and FIDS > FADS contrasts. Thresholded at $P < 0.001$ uncorrected. Orange: yellow + red, purple: blue + red. F female, M male, DDS dog-directed speech, IDS infant-directed speech, ADS adult-directed speech. L left, R right, S superior, I inferior, P posterior, A anterior, mESG mid ectosylvian gyrus, mSSS mid suprasylvian sulcus, c/rSG caudo-rostral transition of the Sylvian gyrus, cSG/TP a region involving the ventralmost part of the caudal sylvian gyrus and the temporal pole. $N = 19$.

change; Spectral Center of Gravity; Harmonics-to-Noise Ratio; jitter and call length; see Methods for details). These results are in line with previously reported findings on infant-, dog- and adult-directed speech prosody[10,13]. Then, in a series of parametric modulatory analysis, we tested how these acoustic features modulate brain responses at the whole brain and in selected small volumes (ROIs: L c/rSG, L cSG/TP, Table 2). Both levels revealed positive modulatory effects of F0 mean and F0 variance on L c/rSG responses to auditory stimuli (Table 2, Fig. 3). Results of the small volume corrected analysis also showed positive association between F0 mean and L cSG/TP responses (Table 2, Fig. 3). No modulatory effect of other acoustic parameters has been found either in the whole brain or in the small-volume analyses (Table 2).

## Discussion

In the present paper, we used non-invasive fMRI to investigate dogs' brain responses to addressee-specific (i.e., dog-, infant- and adult-directed) prosody of female and male speakers. (i) We found that a non-primary auditory brain region of dogs, the left caudal/rostral Sylvian gyrus, extending to the temporal pole exhibited greater activity to exaggerated (dog- and infant-directed) than to adult-directed prosody, especially in women's speech. (ii) Parametric modulation analyses revealed that speech with higher and more varying fundamental frequency (F0, perceived as pitch) elicited greater responses in these non-primary auditory cortical regions of dogs.

The present findings demonstrate the involvement of the ventralmost portion of the dog auditory cortex in processing addressee-specific prosody in speech. These secondary auditory regions have already been implicated in voice processing. Specifically, the temporal pole (TP) in dogs has been shown to respond stronger to conspecific vocalizations than to other sounds[31], and to the owner's voice than to a familiar human's voice[6]. In contrast, modulatory effects of emotional prosody have been reported in

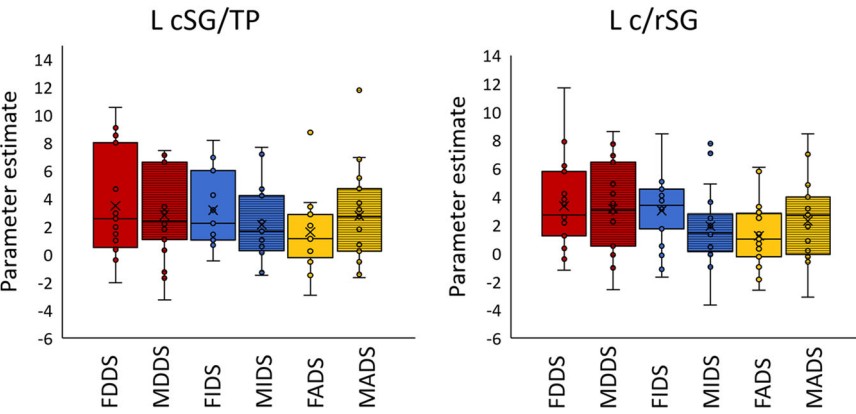

**Fig. 2 Neural activity measured in the two ROIs.** Bar graphs showing response profiles (auditory condition > Sil) in selected ROIs (L c/rSG: transition of the left caudal and rostral sylvian gyrus, L cSG/TP: a region involving the ventralmost part of the caudal sylvian gyrus and the temporal pole). Error bars indicate 95% confidence interval. F female, M male, DDS dog-directed speech, IDS infant-directed speech, ADS adult-directed speech. Upper and lower lines of the rectangle represent the second and third quartiles, the vertical line inside indicates median value, X indicates average value. Horizontal lines either side of the rectangle show lower and upper quartiles. $N = 19$.

**Table 2 Parametric modulatory-effects of acoustic features.**

| Effect | Brain region | x | y | z | Cluster size | T12 | P$_{FWE}$ cluster | Search space |
|---|---|---|---|---|---|---|---|---|
| Positive modulatory effects | | | | | | | | |
| F0 mean | L c/rSG | −24 | −14 | 8 | 43 | 8.141 | <0.001 | Whole brain |
| | L c/rSG | −24 | −14 | 8 | 24 | 8.141 | <0.001 | L c/rSG |
| | L cSG/TP | −20 | −6 | −4 | 16 | 5.354 | <0.001 | L cSG-TP |
| F0 variance | L c/rSG | −22 | −18 | 8 | 14 | 5.702 | 0.043 | Whole brain |
| | L c/rSG | −22 | −18 | 8 | 13 | 5.605 | <0.001 | L c/rSG |
| F0 change | n.s. | | | | | | | |
| HNR mean | n.s. | | | | | | | |
| SCG mean | n.s. | | | | | | | |
| Jitter | n.s. | | | | | | | |
| Call length | n.s. | | | | | | | |

All negative modulatory effects were non-significant
Threshold for reporting for all contrasts was P < 0.001 and cluster-corrected P < 0.05 for FWE, clusters of minimum 3 voxels in both whole-brain or small-volume cluster-corrected levels. Only the strongest peaks are reported.
F0 fundamental frequency, HNR harmonic-to-noise ratio, SCG spectral center of gravity, L left, c/rSG caudo-rostral transition of the Sylvian gyrus, cSG/TP a region involving the ventralmost part of the caudal sylvian gyrus and the temporal pole, n.s. no significant clusters.

near-primary rather than in secondary auditory cortices[2,5,31,32]. Taken together, these results suggest that prosody-sensitivity in the anterior (ventralmost) portion of the dog temporal cortex, including the temporal pole, may reflect vocal stimuli's communicative relevance and attention-grabbing ability rather than their emotional valence.

While fMRI studies on preverbal infants are rare due to methodological difficulties, a similar effect of addressee-specific prosody has been shown in 4- to 13-month-olds with NIRS and ERP. Infant-directed speech (IDS) in these studies evoked more intense neural activation in the left or bilateral temporal areas than adult-directed speech (ADS)[23,25]. Our results indicate that these areas in the dog brain, similarly to infants, show sensitivity for speech directed to them. The lack of differences in the responses to DDS and IDS in dog brains supports the analogue function of the two speech styles and is probably due to their similar acoustic characteristics (e.g.[10]). We revealed functional similarities between dogs' and infants' addressee-specific prosody processing at a neural level. This similarity and the applicability of dogs in non-invasive fMRI studies[1] make the family dog a promising animal model of preverbal infants.

The findings of increased sensitivity to exaggerated prosody of female compared to male speech in dogs, and that their brains differentiated IDS from ADS only when produced by a female

speaker parallels recent NIRS reports in infants[22]. Female voice sensitivity of infants is usually explained by ancient sensitivity to conspecific female voices or by extended intrauterine and postnatal exposure to the mother's voice[19]. Others suggest that female voices in general have greater potential to evoke infant's attention and responsiveness from the infant because women have typically higher-pitched voices than men due to anatomical differences between males and females (for a review see ref.[20]). Although these accounts are difficult to disentangle in infants, the fact that we see a similar pattern in dogs can help in deciding between possible explanations. First, responsiveness of dog brains to exaggerated prosody of female speech cannot be explained by ancient sensitivity to conspecific signals. Second, the ear canal is closed until about 3 weeks of age in dogs (e.g.[33]), therefore we can exclude the effect of intrauterine and early postnatal exposure to female voice on dogs' preference for female addressee-specific prosody.

Two explanations can be given for how such gender- and addressee-dependent prosody sensitivity may have emerged in the dog brain. (i) It can be originated from an ancient and universal sensitivity in the animal kingdom that is linked to greater attention toward sounds with higher pitch and greater F0 variability presented typically in more aroused vocalizations (e.g.[34,35]). (ii) It can be specific to animals that are living and developing in

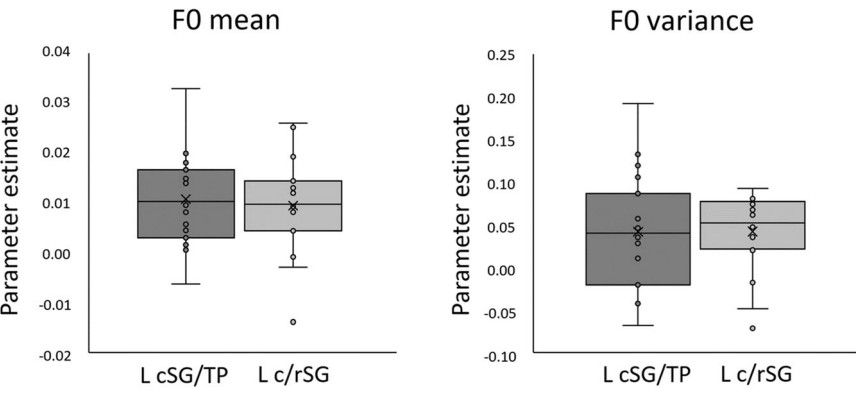

**Fig. 3 Neural activity in response to fundamental frequency mean and range measured in the two ROIs.** Bar graphs showing neural activity in selected ROIs (L c/rSG: transition of the left caudal and rostral sylvian gyrus, L cSG/TP: a region involving the ventralmost part of the caudal sylvian gyrus and the temporal pole). F0 fundamental frequency. Upper and lower lines of the rectangle represent the second and third quartiles, the vertical line inside indicates median value, X indicates average value. Horizontal lines either side of the rectangle show lower and upper quartiles. $N = 19$.

an anthropogenic environment, where speech is part of natural environmental stimuli (e.g.[36,37]). It is widely accepted that dogs during the process of domestication have been selected for traits that enable them to be sensitive to human non-verbal communicative signals (e.g.[38]). If we extend this idea to verbal communication in general and to sensitization to addressee-specific prosody in particular, we can assume that individuals sensitive to dog-directed speech prosody were more likely to stay close to humans and pay attention to their vocal cues. There is evidence that hand-raised wolves are more willing to attend to low-pitched intonations, at the same time higher frequencies in speech are more likely to call the attention of similarly raised dogs[39]. This further supports the notion that such neural sensitivity to relevant speech prosody have developed during the course of domestication. However, further studies with other domesticated species (i.e., horses, cats, etc.) are necessary to study this question. Importantly, however, the potential developmental effects of the acoustic environment on prosody perception must also be considered. During their ontogeny, dogs' experience with human speech is biased to their owner's voice and to their individual acoustic characteristics. If so, individual auditory experiences (developmental factor) can at least partly account for the results of the present study. Most probably, these two mechanisms, the ancient preference, and ontogenetic experiences act together. Higher-pitched voices have the potential to call and maintain dogs' attention, therefore owners use dog-directed prosody, as a feasible communicative style with canine companions. In parallel, owners are rewarding dogs on a daily basis while using dog-directed prosody which amplifies this neural sensitivity during dogs' life span.

The finding that neural sensitivity to F0 mean and F0 variance plays a major role in prosody processing in dogs is in line with previous reports[2,31,32]. Great F0 mean and F0 variability in IDS are considered as important acoustic parameters that drive newborns' and infants' attention toward the speaking mother (e.g.[40–42]). The present neural findings suggest that F0 mean and F0 variance are key acoustic parameters to engage a non-speaking partner (i.e., dog or preverbal infant) during vocal interactions.

An interesting question is whether the found neural sensitivity to F0 mean and variance, is solely responsible for the previously demonstrated attention-getting ability of exaggerated prosody (i.e., DDS and IDS[9,30]). In other words, would non-speech stimuli with the same F0 mean and variance have a similar effect in dog brains? Behavioral studies on dogs revealed that acoustics has a key role in making DDS attention-grabbing. Specifically, both

increased F0 mean (e.g.[9,43]) and greater F0 variance[30] in DDS have been shown to enhance dogs' attention. Even though DDS may have non-acoustic components (e.g., specific lexical content) that also contribute to it being attention-grabbing, the acoustic properties of DDS alone, are also sufficient to attract dogs' attention, as shown with non-speech sine-wave sounds with matched mean F0[30]. These results are in line with studies on the attention-grabbing properties of IDS in preverbal infants (e.g.[42,44]).

The parametric modulatory analyses reported in this study suggest that dogs' auditory cortical responses to DDS are affected by both F0 mean and F0 variance, but it is a valid question whether these effects reflect (bottom-up) acoustic or (top-down) acoustic-independent (for instance attentional) processes. When investigating the processing of acoustic parameters carried by voice or voice-like stimuli, previous dog fMRI studies revealed negative modulatory effects of F0 mean in lower-level, primary and near-primary auditory regions[31,32]. In the current study, however, we found positive modulatory effects of F0 mean in higher order, non-primary auditory dog brain regions. Both the direction and the localization of the F0 mean-related parametric modulation effects found here indicate that these may reflect the involvement of acoustic-independent processes, for instance increased attention in response to exaggerated prosody presented in DDS/IDS. In support of this, in previous dog fMRI studies activity increase in higher order, non-primary auditory cortical dog brain regions was found in response to acoustic-independent factors, such as lexical meaningfulness[2] and attachment to the speaker[6]. Higher order auditory brain responses are modulated by acoustic-independent factors, such as attention, also in humans[45]. Nevertheless, we emphasize that the present study was not designed or aimed to study this question. Future studies are needed to separately investigate the effects of acoustic and non-acoustic components of DDS and IDS prosody on dogs' attention at behavioral and neural level.

Limitations of the current study should be taken into account. In our sample, we used specifically trained family dogs who are able to voluntarily stay still in the scanner (for detailed training methods see the "Methods" section). Due to the restricted number of potential subjects, we had limited opportunity to control for the owner's gender and the dog's breed, however, we included dogs from various breeds who are interacting with people from both genders on a daily basis to extend the generalizability of our results (see the "Methods" section for details). At the same time, we emphasize that our sample is not suitable

for systematic investigations on the inherited and/or learnt factors, nor the influence of dog breed that potentially can have an effect on the results. Despite that we found no association between dogs' head shape and their cerebral responses, studies support that diverse dog breed types, due to recent selection[46] vary in their efficiency in processing human communication cues[47,48]. Future studies with different dog breed types (varying broadly in genetic ancestry, breed function and head shape) may be informative with regard to domestication- and recent selection-related effects.

In sum, to the best of our knowledge, our study provided the first neural evidence in a non-human animal for gender- and addressee-dependent neural sensitivity, and neural sensitivity in non-primary auditory regions for F0 mean and variance of addressee-specific speech prosody . Our results provided important underlying ostension-decoding mechanisms in dogs at a neural level that could have a major role in engaging a canine partner during communicative interactions. Here, we propose that dog- and infant-directed speech prosody, as a behaviorally relevant, complex non-conspecific vocal signal has the potential to evoke neural sensitivity in dogs and highlight the importance of future comparative studies on the neural mechanism of addressee-specific prosody processing.

## Methods

**Ethical statement**. The study has complied with all relevant ethical regulations for animal testing. Ethical approval for this study was obtained from the *National Scientific Ethical Committee on Animal Experimentation* (PEI/001/1057-6/2015). Owners volunteered with their dogs to participate in the study, did not get any monetary compensation and gave a written informed consent.

**Subjects**. 19 adult family dogs (*Canis familiaris*) participated in the current study. To increase generalizability of the results to all dogs, our subjects were selected from various breeds (5 Golden retrievers, 3 Border collies, 3 Cocker spaniels, 1 Australian shepherd, 1 Chinese crested, 1 Cairn terrier, 1 Labrador retriever and 4 mongrels; aged 2–10 years, mean = 6, SD = 2.9; 11 males and 8 females). All subjects had daily interactions with both females and males, and the majority of them (16/19) lived in the same household with humans of both genders. While prior experiences with individuals of different genders can impact dogs' responses to male and female voices[27], in this study there was minimal variability with this respect.

**Dog training**. Prior to the experiment, dogs completed a special training including elements of conditioning and social learning, utilizing positive reinforcement to teach dogs to remain still during the scanning process (see also refs. [2,31]). The training program was a step-by-step procedure in which the dogs were gradually prepared for the awake brain imaging. This involved teaching the dogs to maintain the required position on a table, get used to earphones and strips/coil, and become accustomed to the noise and vibration of the scanner. The dogs were rewarded with food, verbal praise, and petting for exhibiting the desired behavior. In the beginning of the training procedure, some aspects of the "Model/Rival" training method[49] were adopted. In this method, novice dogs were allowed to off-leash observe the performance of experienced model dogs. When the model dog was rewarded by the trainer, as well as by both the model and novice dogs' owners, the novice dog was ignored. This technique highly increased novice dogs' motivation to learn the tasks. No physical restrictions were applied to keep the dogs in the desired position, and dogs could leave the scanner tube at any time.

**Acoustic stimuli**. Dogs were presented with a set of auditory stimuli which included DDS, IDS, ADS (24 for each condition) and with silence (Sil). Auditory stimuli were a subsample of our previously recorded larger sample of DDS, IDS, and ADS[10]. Each stimuli contained full sentences recorded from both female and male speakers unfamiliar to the dogs (N = 12–12) during natural positive interactions with their own babies (IDS) and family dogs (DDS) as well as an adult experimenter (ADS, see ref. [10] for details). From one speaker only one passage has been selected in ADS, DDS and IDS. The stimuli were carefully selected to have the same length on average of IDS, DDS and ADS conditions ($F_{2,69} = 0.025$, $p = 0.98$).

All stimuli have been normalized and equalized (67 dB) with PRAAT program[50] (v 6.1.12). This procedure had no effect on the spectrotemporal parameters of the stimuli. We applied PRAAT program for acoustic analysis in order to analyze fundamental frequency (F0) mean, variance, change; Harmonic-

to-noise ratio (HNR), Spectral Center of gravity (SCG, i.e., spectral centroid), jitter and call length (Table 3).

We used Generalized Linear Mixed Model (GLMM) with Gaussian error distribution to investigate the effect of addressee (ADS, DDS, IDS) and speaker's gender (female, male) on the aforementioned acoustic parameters (IBM SPSS 21). To control for repeated measures, the speaker's ID has been included as a random grouping factor. All tests were two-tailed and the α value was set at 0.05. Bonferroni correction was applied in all post-hoc comparisons. Non-significant interactions and main effects were removed from the model in a stepwise manner (backward elimination technique). Results were in line with previous studies on the acoustic features of DDS, IDS and ADS (see Table 3, Fig. 4, e.g.[8–10,20]).

**Design and procedure**. Four main conditions were used: 1. ADS, 2. DDS 3. IDS, and 4. Silence which was used as a baseline. Stimulus blocks were presented in silent gaps between 2 s long volume acquisitions. Silent gaps were 8 sec long. Each stimulus was positioned in the middle of the silent gap. Stimulus onsets were placed 0.1–1 s (mean = 0.65) after the beginning of the silent gap (Fig. 5). The fMRI test was split into three approximately 7-min-long runs each consisting of 8 repetitions of every condition (including silence), resulting in 32 stimulus presentation blocks per run. In all three runs stimuli of ADS, DDS, and IDS conditions were also counterbalanced to speakers' gender (i.e., in one run 4 stimuli were used from male and 4 from female speakers in each condition) and resulted in 7 conditions in total (referred to as types of stimuli): female (F) DDS/IDS/ADS, male (M) DDS/IDS/ADS and silence. One speaker's voice was used only once per run. The run order was counterbalanced across dogs.

Stimuli were controlled using Matlab (version 9.1) Psychophysics Toolbox 3[51]. During scanning, stimulus presentation and data acquisition were synchronized by a TTL trigger pulse. Stimuli were delivered binaurally through MRI-compatible sound-attenuating headphones that were suitable to cover the ears of the dogs, thus it was suitable for protecting dogs from loud scanner noises as well.

**Data acquisition**. MRI measurements were taken at the Brain Imaging Centre of the Research Centre for Natural Sciences, Budapest, Hungary, on a 3 T Siemens MAGNETOM Prisma syngo MR D13D with a single loop coil. Spatial resolution (i.e., voxel size) was 2.0 × 2.0 × 2.0 mm.

For the functional measurements, a single-shot gradient-echo planar imaging (EPI) sequence acquired the volumes of the entire brain (31, 2.0 mm thick coronal slices in a R ≫ L sequence; TE: 30.0 ms; TR: 10,000 ms; flip angle: 90°; acquisition matrix: 64 × 64). We used 32 volumes in one run, 8 per condition. A T1-weighted anatomical template brain image was taken on a 3 T whole body scanner with a Philips SENSE Flex Medium coil[52].

Dogs were trained to lie motionless for the whole duration of the run. We applied no restriction during the scanning, therefore they could leave the setting anytime by withdrawing their head (see ref. [31]). Dogs were tested with minimum one, maximum two runs per day. We continued data collection until all dogs had 3 successful runs (Dog07 and Dog16 completed only one run, because their owners were not available anymore). Overall motion/rotation threshold was 3 mm and 3° for each direction. If head movements exceeded this threshold the affected and all following volumes of the scan were excluded (55 from 57 functional runs were not affected by these exclusion criteria: volume 26–32 of a run of Dog08 and volume 27–32 of a run of Dog11 were excluded). The average of maximal movements per dog was below 1.6 mm for each translation direction, and below 1.2 degree for each rotation direction. The average scan-to-scan movement of dogs was 0.08 mm.

**Data analysis**. Image preprocessing and statistical analysis were performed using MATLAB R2016b and SPM12. As a brain template for analyses we used the anatomical image of a golden retriever's brain[52]. First, we realigned functional images of the 3 runs in case of each dog. Second, to correct for the different orientation of human and dog heads in the scanner, we manually reoriented each mean-functional and realigned-functional files. Third, we transformed mean-functional images for each dog to the template space via Amira 3D software platform. Fourth, we made a second normalization in spm12, in which we added the normalized-mean-functional files (the ones transformed in Amira 3D) as a template, the original non-normalized-mean-functional files as a source image and we applied the transformation matrix between the two previous images for all realigned-functional files, that resulted all images to be in a shared space. Finally, we convolved with an isotropic 3-D Gaussian kernel (FWHM = 4 mm) for spatial filtering of all normalized-functional files. We centered all images around the Commissura rostralis.

**Statistics and reproducibility**. General Linear Model and statistical parametric mapping were used for data analyses. Our model was built with condition regressors for the 7 conditions (FADS, FDDS, FIDS, MADS, MDDS, MIDS, SIL modeled as a 6.5 long blocks) and for the 3 runs. We also added potential movement artifacts as regressors, resulting from the realignment step. To filter out the potential muscle-movement artifacts, hailed from the tight muscle-tissue around the dogs' head, we used a whole-brain inclusive mask in the individual-level analyses. Single-subject fixed effect analyses were followed by whole-volume random effects analyses on the group level. We searched for effects of addressee

**Table 3 Acoustic analysis of stimuli.**

|  | F0 mean | F0var | F0chg | HNR | SCG | jitter | CL |
|---|---|---|---|---|---|---|---|
| Gender | $F_{1,1113} = 83.31$, $p < 0.001$ | $F_{1,1115} = 18.15$, $p < 0.001$ | $F_{1,1115} = 12.57$, $p < 0.001$ | n.s. | n.s. | $F_{1,1115} = 56.4$, $p < 0.001$ | n.s. |
| Addressee | $F_{2,1113} = 116.52$, $p < 0.001$ | $F_{2,1115} = 16.17$, $p < 0.001$ | $F_{2,1115} = 12.97$, $p < 0.001$ | $F_{2,1113} = 4.75$, $p = 0.009$ | $F_{2,1115} = 21.23$, $p < 0.001$ | n.s. | n.s. |
| Gender× Addressee | $F_{2,1113} = 12.99$, $p < 0.001$ | n.s. | n.s. | $F_{2,1113} = 8.53$, $p < 0.001$ | n.s. | n.s. | $F_{2,1113} = 3.88$, $p = 0.02$ |
| F > M | $p < 0.001$ | $p < 0.001$ | $p < 0.001$ | n.s. | n.s. | $p < 0.001$ | n.s. |
| FDDS > MDDS | $p < 0.001$ | $p = 0.001$ | $p = 0.017$ | n.s. | n.s. | $p < 0.001$ | n.s. |
| FIDS > MIDS | $p < 0.001$ | $p < 0.001$ | $p < 0.001$ | n.s. | n.s. | $p < 0.001$ | $p = 0.025$ |
| FADS > MADS | $p < 0.001$ | $p = 0.046$ | n.s. | n.s. | n.s. | $p < 0.001$ | $p = 0.037$ |
| DDS > ADS | $p < 0.001$ | $p < 0.001$ | $p < 0.001$ | n.s. | $p < 0.001$ | n.s. | n.s. |
| DDS > IDS | $p < 0.001$ | n.s. | n.s. | n.s. | $p < 0.001$ | n.s. | n.s. |
| IDS > ADS | $p < 0.001$ | $p < 0.001$ | $p < 0.001$ | $p = 0.008$ | $p = 0.045$ | n.s. | n.s. |
| FDDS > FADS | $p < 0.001$ | $p < 0.001$ | $p = 0.001$ | $p < 0.001$ | $p = 0.006$ | n.s. | n.s. |
| FDDS > FIDS | $p = 0.003$ | n.s. | n.s. | n.s. | $p < 0.001$ | n.s. | n.s. |
| FIDS > FADS | $p < 0.001$ | $p < 0.001$ | $p < 0.001$ | $p < 0.001$ | n.s. | n.s. | n.s. |
| MDDS > MADS | $p < 0.001$ | n.s. | $p = 0.040$ | n.s. | $p = 0.003$ | n.s. | n.s. |
| MDDS > MIDS | $p < 0.001$ | n.s. | n.s. | $p = 0.04$ | $p < 0.001$ | n.s. | $p = 0.006$ |
| MIDS > MADS | $p < 0.001$ | $p = 0.039$ | n.s. | n.s. | n.s. | n.s. | n.s. |

*F0* Fundamental frequency, *var.* variance, *chg* change, *HNR* Harmonics to noise ratio (mean), *SCG* Spectral Center of gravity (mean), *CL* call length (mean), *ADS* adult-directed speech, *DDS* dog-directed speech, *IDS* infant-directed speech, *F* female, *M* male, n.s.: $p > 0.05$.

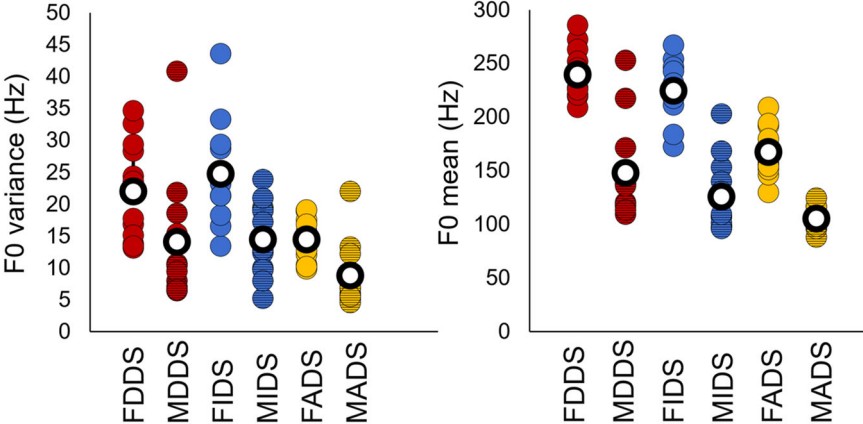

**Fig. 4 Acoustic features of the fMRI stimuli.** Fundamental frequency (F0) mean and variance of the acoustic stimuli. F female, M male, DDS dog-directed speech, IDS infant-directed speech, ADS adult-directed speech, Hz hertz. Number of stimuli per auditory condition: 12. Total number of stimuli: 72.

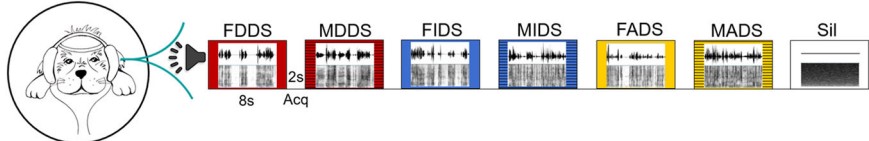

**Fig. 5 Illustration of the data acquisition and design.** This figure shows a dog lying in the scanner and listening to fMRI stimuli, and the sparse scanning design. Stimuli were presented within 8-second-long scanning gaps between 2-second-long volume acquisitions. Conditions were randomized across subjects. F female, M male, DDS dog-directed speech, IDS infant-directed speech, ADS adult-directed speech.

(ADS, DDS and IDS), speaker's gender (F, M) and addressee × gender interaction within the whole brain.

To test whether the obtained results depend on dogs' head shape (cephalic index) we added it as a covariate in a follow-up group level model. We used the contrasts resulting in significant clusters in our first group level model (Table 1). Dogs' brain-based cephalic index is suitable for measuring the impact of breeding on the structure of a dog's brain[53], relevant in the cerebral processing of human auditory cues[7] and related to dogs' human communicative cue-reading abilities (cf[54,55]). Cephalic index was calculated from brain width and length based on the MR anatomical images of each dog (for details see ref.[56]).

In order to identify acoustic parameters that serve as a basis for addressee- and gender-specific neural sensitivity, we performed a series of parametric modulatory analyses adding fundamental frequency (F0) mean, -variance, -change, Harmonic-to-noise ratio (HNR), Spectral Center of gravity (SCG, i.e., spectral centroid), jitter and call length as possible modulatory factors of auditory brain responses. Auditory stimuli were not divided into conditions here. This analysis was performed at both whole brain and small volume corrected levels. For small-volume analysis we defined two 6-mm-radius ROIs where addressee and/or addressee by speaker gender effects were found (i.e. L c/rSG and L cSG/TP) via the marsbar toolbox of SPM. In order to determine the center of the ROIs we averaged the coordinates of the resulted peaks in DDS > ADS, FDDS > FADS, FIDS > FADS, (FDDS + FIDS) > FADS, and (FDD + FID > FAD) > (MDD + MID > MAD) contrasts (Table 1).

**Reporting summary**. Further information on research design is available in the Nature Portfolio Reporting Summary linked to this article.

## Data availability

All data are available on Mendeley Data at https://doi.org/10.17632/76x6zp4j46.1[57]. Source data underlying figures are presented in Supplementary Data 1.

## Code availability

All the codes supporting this study are available on Mendeley Data at https://doi.org/10.17632/76x6zp4j46.1[57].

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

## Acknowledgements

The authors wish to thank Krisztina Hegedűs-Kovács, Borbála Ferenczy, Bernadett Miklósi, Rita Báji, and Dóra Szabó for their assistance in dog training and data collection, Tamás Faragó for his contribution to stimulus preparation and acoustic analysis. The study was supported by Hungarian Scientific Research Fund (NKFIH grant no. FK142968, FK128242, K132372) and by the Hungarian Brain Research Program (HBRP) 3.0 NAP. A.Gá. was supported by the New National Excellence Program of the Ministry for Culture and Innovation from the source of the National Research, Development and Innovation Fund (ÚNKP-22-4-II-ELTE-319). K.C., A.A., and A.Gá. were supported by the European Research Council (ERC, 680040, 950159). A.K. was supported by the János Bolyai Research Scholarship of the Hungarian Academy of Sciences. M.G. was supported by ELKH-ELTE Comparative Ethology Research Group (grant no. F01/031).

## Author contributions

A. Gergely and A. Gábor: Conceptualization, investigation, writing—original draft. M.G.: Investigation, writing—review and editing. A.K. and K.C.: Writing—review and editing. J.T. and A.A.: Conceptualization, supervision, funding, writing—review and editing.

## Funding

## Competing interests

The authors declare no competing interests.
