## [Peer Review File · Communications Biology]

Reviewers' comments:

Reviewer #1 (Remarks to the Author):

I have only a few issues with this otherwise exciting and original study.

1) Difference DDS-IDS. Acoustical parameters shown suggest some differences between IDS and DDS in terms of f_0 ; why has the contrast IDS vs DDS not been examined in the cerebral analysis? There seems to be some interesting differences particularly for male speakers.

2) The 'modulatory' analysis shown in Fig 3B is hard to understand. What is the motivation for these 4 f_0 categories? Why not the same type of analysis for f_0 range for instance? Also, the 4 windows chosen may have the same width in Hz, but given the logarithmic relationship between f_0 and pitch, a 50Hz window at low f_0 s spans a much larger pitch interval (close to one octave) than at higher frequencies where the 50Hz correspond to about a quarter octave... Moreover, these 4 windows are not really equilibrated in terms of number of stimuli (Table 5). How could this affect the analysis? In addition, this modulatory analysis is described after Figure 3, which already includes some of its results.

3) The introduction mentions the 'attention getting' component of IDS and DDS; but are the cerebral differences related to these differences, or purely to acoustical differences? In other words, would we see the same effect by using nonspeech stimuli (e.g. music) with similar acoustical differences?

4) Minor: Centre of Gravity: you mean spectral centroid?

Reviewer #2 (Remarks to the Author):

Congratulations to the authors of this manuscript for a most interesting and well constructed study. It was very interesting to read, and has obvious life applicability and probably broader applications in future as well.

Major Claims:

the major claims in this work are that dogs resemble human infants in their bias towards exaggerated prosody such as in typical female speech, they claim that the functional neurological response to female prosody significantly activates secondary auditory cortical areas (temporal pole, ventral part of the caudal Sylvian gyrus; caudo-rostral Sylvian gyrus) than male prosody; and further they claim that both dog and in fact directed speech creates more activation within these regions, significantly more than adult-directed speech.

Novelty and Interest:

These findings are certainly novel and very interesting, fMRI recordings of brain activity in dogs certainly presents rare information not so readily available, especially in non-human creatures. There is great potential for other neuroscientists to learn from these findings and perhaps expand to learn about other animals' neural responses to such different types of speech.

The work is convincing

Influence on thinking in the field:

yes, this work will influence thinking in the field, moreover is very relevant given the commonality of the domestication of dogs.

Questions/Concerns:

1. Given that authors could not control for female/male ownership of subjects, nor previous experiences with either for the subject, how might controlling for these aspects change the results? In other words, were there any observations of difference between the male-owned v/s female owned subjects in their responses to the various speech types?
2. Did authors look into stratifying the subjects by dog breed? could breed be a factor in the extent of its neural response at all?

STATS:

The statistical analyses conducted here were thorough and enough to address the various research questions posed by the study. They were all appropriate.

REPRODUCIBILITY:

Enough information is given here such that another researcher would be able to reproduce the study. It would have been good to read on the dog training process, however it is appropriate that it was left out and rather referenced.

CORRECTIONS:

Please correct the below minor grammatical errors:

314 Dogs were presented with a set of auditory stimuli (Figure 4) which included DDS, IDS, ADS –
CORRECTION: add the missing comma before "ADS" in this sentence.

362 ...in each condition) and resulted in 7 conditions in total (referred to as stimuli types): female
(F) – CORRECTION: add the missing s to "7 conditions"

196 perceived as pitch) elicited greater responses in these auditory regions of dogs. (iv) Finally,
CORRECTION: roman numeral here should be (iii) and not (vi)

242-243 Great F0 variability in IDS is considered as an important acoustic parameter that drives
newborns' and infants' attention...CORRECTION: add |"is" after "in ADS"

Response to Reviewers

Reviewer #1

I have only a few issues with this otherwise exciting and original study.

We are thankful to the Reviewer for their supportive comments. We believe that the revisions we made in response to their suggestions improved the clarity of the manuscript. Below we elaborate on each comment in turn.

1) Difference DDS-IDS. Acoustical parameters shown suggest some differences between IDS and DDS in terms of f0; why has the contrast IDS vs DDS not been examined in the cerebral analysis? There seems to be some interesting differences particularly for male speakers.

Although we performed the requested contrasts in the originally submitted manuscript already, we agree with this Reviewer that the description and tables were not clear enough. Following this concern, we listed all tested contrasts one by one in the caption of Table 1, and we expanded the MS with their further discussion (Line 112-118 and Line 202-206). None of the newly listed contrasts resulted in significant clusters.

2) The ‘modulatory’ analysis shown in Fig 3B is hard to understand. What is the motivation for these 4 f0 categories? Why not the same type of analysis for f0 range for instance? Also, the 4 windows chosen may have the same width in Hz. but given the logarithmic relationship between f0 and pitch, a 50Hz window at low f0s spans a much larger pitch interval (close to one octave) than at higher frequencies where the 50Hz correspond to about a quarter octave... Moreover, these 4 windows are not really equilibrated in terms of number of stimuli (Table 5). How could this affect the analysis? In addition, this modulatory analysis is described after Figure 3. which already includes some of its results.

We are grateful to the Reviewer for raising these valid questions regarding the follow-up analysis on the F0 mean categories. The original aim of this analysis was to investigate the modulatory effect of the F0 mean in more detail. At the same time, we agree with the Reviewer that these categories were not perceptually equally distributed, and different and low numbers of stimuli per category made this analysis less reliable. We attempted to create more equal categories based on F0 mean and F0 variance, but our study was not initially designed for such detailed follow-up analyses. Therefore, after careful consideration, we have decided to remove this follow-up analysis from the revised manuscript and focus more on discussing the robust results presented in Table 2 and Figure 3. We believe that this decision did not significantly impact the main conclusions.

Figure 3. Neural activity in response to fundamental frequency mean and range measured in the two ROIs

Bar graphs showing neural activity in selected ROIs (L c/rSG: transition of the left caudal and rostral sylvian gyrus, L cSG/TP: a region involving the ventralmost part of the caudal sylvian gyrus and the temporal pole). F0: fundamental frequency.

3) The introduction mentions the ‘attention getting’ component of IDS and DDS; but are the cerebral differences related to these differences, or purely to acoustical differences? In other words, would we see the same effect by using nonspeech stimuli (e.g. music) with similar acoustical differences?

We thank this Reviewer for posing this thought-provoking question. Behavioural studies on dogs revealed that acoustics has a key role in making DDS attention-grabbing. Specifically, both increased F0 mean (e.g. Ben-Aderet et al 2017, Jeannin et al 2017) and greater F0 variance (Gergely et al 2021) in DDS have been shown to enhance dogs’ attention. And even though DDS may have non-acoustic components (e.g. specific lexical content) that contribute to it being attention-grabbing, the acoustic properties of DDS alone, are also sufficient to attract dogs’ attention, as shown with non-speech sine-wave sounds with matched F0 (Gergely et al 2021). These results are in line with studies on the attention-grabbing properties of IDS in preverbal infants (e.g. Fernald & Kuhl 1987, Cooper & Aslin 1990).

The parametric modulatory analyses performed in this study suggest that dogs’ auditory cortical responses to DDS are affected by both F0 mean and F0 variance, but it is a valid question whether these effects reflect (bottom-up) acoustic or (top-down) acoustic-independent (for instance attentional) processes. When investigating the processing of acoustic parameters carried by voice or voice-like stimuli, previous dog fMRI studies revealed negative modulatory effects of F0 mean in lower-level, primary and near-primary auditory regions (Andics et al 2014, Bálint et al 2023). In the current study, however, we found positive modulatory effects of F0 mean in higher order, non-primary auditory dog brain regions. Both the direction and the localization of the F0 mean-related parametric modulation effects found here indicate that these may reflect the involvement of acoustic-independent processes, for instance increased attention in response to exaggerated prosody presented in DDS/IDS. In support of this, in previous dog fMRI studies activity increase in higher order, non-primary auditory cortical dog brain regions

was found in response to acoustic-independent factors, such as lexical meaningfulness (Andics et al 2016) and attachment to the speaker (Gábor et al 2021). Higher order auditory brain responses are modulated by acoustic-independent factors, such as attention, also in humans (Fritz et al 2007). Nevertheless, we emphasise that the present study was not designed or aimed to study this question. Future studies are needed to separately investigate the effects of acoustic and non-acoustic components of DDS and IDS prosody on dogs' attention at behavioural and neural level. We expanded the MS with further discussion of this issue, considering the results of other studies (Line 254-264).

4) Minor: Centre of Gravity: you mean spectral centroid?

Yes. Centre of Gravity (COG) or spectral centroid referred to the centre of the spectrum. Since the Praat software we used refers to this spectral parameter as COG, we adopted this nomenclature in our manuscript, similarly to our previous publications (Gergely et al 2017, Andics & Miklósi 2018). However, we acknowledge that the term "center of gravity" is ambiguous. To address this, we added the word "spectral" to the term, resulting in "Spectral Centre of Gravity" and we modified its abbreviation to SCG. Additionally, we included the term "spectral centroid" in the revised manuscript to further clarify SCG (Lines 151, 166, 351-352, 364, 453).

Reviewer #2

Congratulations to the authors of this manuscript for a most interesting and well constructed study. It was very interesting to read. and has obvious life applicability and probably broader applications in future as well.

Major Claims:

The major claims in this work are that dogs resemble human infants in their bias towards exaggerated prosody such as in typical female speech, they claim that the functional neurological response to female prosody significantly activates secondary auditory cortical areas (temporal pole. ventral part of the caudal Sylvian gyrus; caudo-rostral Sylvian gyrus) than male prosody; and further they claim that both dog and in fact directed speech creates more activation within these regions. significantly more than adult-directed speech.

Novelty and Interest:

These findings are certainly novel and very interesting. fMRI recordings of brain activity in dogs certainly presents rare information not so readily available. especially in non-human

creatures. There is great potential for other neuroscientists to learn from these findings and perhaps expand to learn about other animals' neural responses to such different types of speech.

The work is convincing

Influence on thinking in the field:

Yes, this work will influence thinking in the field, moreover is very relevant given the commonality of the domestication of dogs.

We are grateful to the Reviewer for the appreciative words and constructive comments. We believe that our revisions following their suggestions improved the quality of the manuscript. Below, we elaborate on each concern in turn.

Questions/Concerns:

1. Given that authors could not control for female/male ownership of subjects. nor previous experiences with either for the subject, how might controlling for these aspects change the results? In other words, were there any observations of difference between the male-owned v/s female owned subjects in their responses to the various speech types?

All subjects had daily interactions with both females and males, and the majority of them (16/19) even lived in the same household with humans of both genders. While prior experiences with individuals of different genders can impact dogs' responses to male and female voices (Ratcliffe et al. 2016), in our study this case is unlikely. Whereas it would be indeed interesting to test how owner gender affects dog brain responses, the current design is not suitable to perform such analyses.

To elaborate further on this point, we have included additional details in the Methods section (Lines 318-322), and noted this issue among the limitations of the study (Line 285-291).

2. Did authors look into stratifying the subjects by dog breed? could breed be a factor in the extent of its neural response at all?

While there could be breed differences in the processing of addressee-specific prosody in dogs, here our aim was to make conclusions on dogs in general. Therefore, to increase the generalizability of our data, we involved subjects belonging to numerous different breeds. As we had at most a few individuals per breed, based on this study we cannot make conclusions on dog breed differences specifically.

More generally, dog domestication/breeding effects can be investigated by exploring the influence of genetic ancestry (basal vs. modern breeds), breed function (cooperative vs. independent) and head shape (cephalic index). Our study sample was not suitable to investigate the first two, as most of the tested dogs were from modern, cooperative breeds. We could, however, test whether the obtained results depend on the head shape (cephalic index) that is suitable for measuring the impact of breeding on the structure of a dog's brain (Hecht et al 2019). Dogs' brain-based cephalic index is relevant in the cerebral processing of human auditory cues (Cuaya et al 2021) and it is related to dogs' human communicative cue-reading abilities (cf. Bognár et al 2021, and Gácsi et al 2009). We calculated the cephalic index from brain width and length based on the MR anatomical images of each dog (for details see Bunford et al 2020) and we added it as a covariate in a follow-up group level model. This follow-up analysis using the contrasts revealing significant effects in the previous analysis (see Table 1) revealed no head shape effects either on a whole-brain level, or within the relevant ROIs (L c/rSG, L cSG/TP) thresholded at cluster-corrected $P < 0.05$ for FWE. Note, however, that while the cephalic index varied across the sample, all of our dog subjects were medium-headed (mesocephalic), not reflecting the true head shape variability of the species. Given that diverse dog breed types, due to recent selection (MacLean et al 2019), vary in the processing of human communication cues (Passalacqua et al 2011, Persson et al 2015), future studies with different dog breed types (varying broadly in genetic ancestry, breed function and head shape) may be informative with regard to domestication- and recent selection-related effects.

The method and the results of the newly performed analysis with the cephalic index are described and discussed in the MS (Lines 141-144, 285-296, 443-449).

STATS:

The statistical analyses conducted here were thorough and enough to address the various research questions posed by the study. They were all appropriate.

REPRODUCIBILITY:

Enough information is given here such that another researcher would be able to reproduce the study, it would have been good to read on the dog training process. however it is appropriate that it was left out and rather referenced.

We expanded the Methods with a more detailed description of the training procedure. (Line 324-337)

CORRECTIONS:

Please correct the below minor grammatical errors:

We would like to express our gratitude to this Reviewer for their careful reading of the manuscript. We have made the suggested corrections as per their feedback.

314 Dogs were presented with a set of auditory stimuli (Figure 4) which included DDS. IDS. ADS – CORRECTION: add the missing comma before "ADS" in this sentence.

362 ...in each condition) and resulted in 7 conditions in total (referred to as stimuli types): female (F) – CORRECTION: add the missing s to "7 conditions"

196 perceived as pitch) elicited greater responses in these auditory regions of dogs. (iv) Finally. CORRECTION: roman numeral here should be (iii) and not (vi)

242-243 Great F0 variability in IDS is considered as an important acoustic parameter that drives newborns' and infants' attention...CORRECTION: add |'is" after "in ADS"

SIGNED: Busisiwe C. Maseko

We appreciate the time that both Reviewers have taken to evaluate the manuscript!

References

Andics, A., Gácsi, M., Faragó, T., Kis, A., & Miklósi, Á. (2014). Voice-Sensitive Regions in the Dog and Human Brain Are Revealed by Comparative fMRI. *Current Biology*, 24, 574–578. <https://doi.org/10.1016/j.cub.2014.01.058>

Andics, A., Gábor, A., Gácsi, M., Faragó, T., Szabó, D., & Miklósi, Á. (2016). Neural mechanism for lexical processing in dogs. *Science*, 353(63), 1030–1032. <https://doi.org/10.1126/science.aaf3777>

Andics, A., & Miklósi, Á. (2018). Neural processes of vocal social perception: Dog-human comparative fMRI studies. *Neuroscience and Biobehavioral Reviews*, 85, 54–64. <https://doi.org/10.1016/j.neubiorev.2017.11.017>

Bálint, A., Szabó, Á., Andics, A., & Gácsi, M. (2023). Dog and human neural sensitivity to voicelikeness: A comparative fMRI study. *NeuroImage*, 265, 119791. <https://doi.org/https://doi.org/10.1016/j.neuroimage.2022.119791>

- Ben-Aderet, T., Gallego-Abenza, M., Reby, D., & Mathevon, N. (2017). Dog-directed speech: Why do we use it and do dogs pay attention to it? *Proceedings of the Royal Society B: Biological Sciences*, 284(1846). <https://doi.org/10.1098/rspb.2016.2429>
- Bognár, Z., Szabó, D., Deés, A. & Kubinyi, E. (2021). Shorter headed dogs, visually cooperative breeds, younger and playful dogs form eye contact faster with an unfamiliar human. *Scientific Reports*, 11:9293, <https://doi.org/10.1038/s41598-021-88702-w>
- Bunford, N. et al. (2020) Comparative brain imaging reveals analogous and divergent patterns of species and face sensitivity in humans and dogs. *The Journal of Neuroscience*, 40, 8396–8408. <https://doi.org/10.1523/JNEUROSCI.2800-19.2020>
- Cooper, R. P. & Aslin, R. N. (1990). Preference for Infant-Directed Speech in the First Month after Birth Preference for Infant-directed First Month after Birth. *Child Development*, 61, 1584–1595. <https://doi.org/10.1111/j.1467-8624.1990.tb02885.x>
- Fernald, A., & Kuhl, P. (1987). Acoustic determinants of infant preference for motherese speech. *Infant Behavior and Development*, 10, 279–293. [https://doi.org/10.1016/0163-6383\(87\)90017-8](https://doi.org/10.1016/0163-6383(87)90017-8)
- Fritz, J. B., Elhilali, M., David, S. V. & Shamma, S. A. (2007). Auditory attention - focusing the searchlight on sound. *Current Opinion in Neurobiology*, 17, 437–455. <https://doi.org/10.1016/j.conb.2007.07>
- Gábor, A., Andics, A., Miklósi, Á., Czeibert, K., Carreiro, C., & Gácsi, M. (2021). Social relationship-dependent neural response to speech in dogs. *NeuroImage*, 243. <https://doi.org/10.1016/j.neuroimage.2021.118480>
- Gácsi, M., McGreevy, P., Kara, E. & Miklósi, Á. (2009). Behavioral and Brain Functions Effects of selection for cooperation and attention in dogs. *Behavioral and brain functions*, 8, 1–8. <https://doi.org/10.1186/1744-9081-5-31>
- Gergely, A., Faragó, T., Galambos, Á., & Topál, J. (2017). Differential effects of speech situations on mothers' and fathers' infant-directed and dog-directed speech: An acoustic analysis. *Scientific Reports*, 7(1). <https://doi.org/10.1038/s41598-017-13883-2>
- Gergely, A., Tóth, K., Faragó, T., & Topál, J. (2021). Is it all about the pitch ? Acoustic determinants of dog-directed speech preference in domestic dogs , *Canis familiaris*. *Animal Behaviour*, 176, 167–174. <https://doi.org/10.1016/j.anbehav.2021.04.008>
- Hecht, E. E. et al. (2019). Significant neuroanatomical variation among domestic dog breeds. *The Journal of Neuroscience*, 39, 7748–7758. <https://doi.org/10.1523/JNEUROSCI.0303-19.2019>

Jeannin, S., Gilbert, C., Amy, M., & Leboucher, G. (2017). Pet-directed speech draws adult dogs' attention more efficiently than Adult-directed speech. *Scientific Reports*, 7(1), 1–9. <https://doi.org/10.1038/s41598-017-04671-z>

MacLean, E. L., Snyder-Mackler, N., VonHoldt, B. M. & Serpell, J. A. (2019). Highly heritable and functionally relevant breed differences in dog behaviour. *Proceedings of the Royal Society B: Biological Sciences*, 286. <http://dx.doi.org/10.1098/rspb.2019.0716>

Passalacqua, C. et al. (2011). Human-directed gazing behaviour in puppies and adult dogs, *Canis lupus familiaris*. *Animal Behaviour*, 82, 1043–1050. <http://doi:10.1016/j.anbehav.2011.07.039>

Persson, M. E., Roth, L. S. V., Johnsson, M., Wright, D. & Jensen, P. (2015). Human-directed social behaviour in dogs shows significant heritability. *Genes, Brain and Behavior*, 14, 337–344. <http://doi:10.1111/gbb.12194>

Ratcliffe, V. F., McComb, K. & Reby, D. (2014). Cross-modal discrimination of human gender by domestic dogs. *Animal behaviour*, 91, 127-135. <https://doi.org/10.1016/j.anbehav.2014.03.009>

REVIEWERS' COMMENTS:

Reviewer #1 (Remarks to the Author):

The authors have satisfactorily addressed my comments.

Reviewer #2 (Remarks to the Author):

The evidence presented in the work is enough and appropriate to demonstrate the claims made by the researchers. the work is convincing based on the research reported herein. The statistical analysis done was robust and convincing as well. The revisions made to the manuscript make it clearer and indeed are an improvement to an already excellent work by the researchers herein.